# Unsupervised learning of an efficient short-term memory network

**Pietro Vertechi**      **Wieland Brendel** *      **Christian K. Machens**

Champalimaud Neuroscience Programme
Champalimaud Centre for the Unknown
Lisbon, Portugal
`first.last@neuro.fchampalimaud.org`

## Abstract

Learning in recurrent neural networks has been a topic fraught with difficulties and problems. We here report substantial progress in the unsupervised learning of recurrent networks that can keep track of an input signal. Specifically, we show how these networks can learn to efficiently represent their present and past inputs, based on local learning rules only. Our results are based on several key insights. First, we develop a local learning rule for the recurrent weights whose main aim is to drive the network into a regime where, on average, feedforward signal inputs are canceled by recurrent inputs. We show that this learning rule minimizes a cost function. Second, we develop a local learning rule for the feedforward weights that, based on networks in which recurrent inputs already predict feedforward inputs, further minimizes the cost. Third, we show how the learning rules can be modified such that the network can directly encode non-whitened inputs. Fourth, we show that these learning rules can also be applied to a network that feeds a time-delayed version of the network output back into itself. As a consequence, the network starts to efficiently represent both its signal inputs and their history. We develop our main theory for linear networks, but then sketch how the learning rules could be transferred to balanced, spiking networks.

## 1   Introduction

Many brain circuits are known to maintain information over short periods of time in the firing of their neurons [15]. Such "persistent activity" is likely to arise through reverberation of activity due to recurrent synapses. While many recurrent network models have been designed that remain active after transient stimulation, such as hand-designed attractor networks [21, 14] or randomly generated reservoir networks [10, 13], how neural networks can *learn* to remain active is less well understood.

The problem of learning to remember the input history has mostly been addressed in supervised learning of recurrent networks. The classical approaches are based on backpropagation through time [22, 6]. However, apart from convergence issues, backpropagation through time is not a feasible method for biological systems. More recent work has drawn attention to random recurrent neural networks, which already provide a reservoir of time constants that allow to store and read out memories [10, 13]. Several studies have focused on the question of how to optimize such networks to the task at hand (see [12] for a review), however, the generality of the underlying learning rules is often not fully understood, since many rules are not based on analytical results or convergence proofs.

The unsupervised learning of short-term memory systems, on the other hand, is largely unchartered territory. While there have been several "bottom-up" studies that use biologically realistic learning rules and simulations (see e.g. [11]), we are not aware of any analytical results based on local learning rules.

Here we report substantial progress in following through a normative, "top-down" approach that results in a recurrent neural network with local synaptic plasticity. This network learns how to efficiently remember an input and its history. The learning rules are largely Hebbian or covariance-based, but separate recurrent and feedforward inputs. Based on recent progress in deriving integrate-and-fire neurons from optimality principles [3, 4], we furthermore sketch how an equivalent spiking network with local learning rules could be derived. Our approach generalizes analogous work in the setting of efficient coding of an instantaneous signal, as developed in [16, 19, 23, 4, 1].

## 2 The autoencoder revisited

We start by recapitulating the autoencoder network shown in Fig. 1a. The autoencoder transforms a $K$-dimensional input signal, $\mathbf{x}$, into a set of $N$ firing rates, $\mathbf{r}$, while obeying two constraints. First, the input signal should be reconstructable from the output firing rates. A common assumption is that the input can be recovered through a linear decoder, $\mathbf{D}$, so that

$$\mathbf{x} \approx \hat{\mathbf{x}} = \mathbf{Dr}. \tag{1}$$

Second, the output firing rates, $\mathbf{r}$, should provide an optimal or efficient representation of the input signals. This optimality can be measured by defining a cost $C(\mathbf{r})$ for the representation $\mathbf{r}$. For simplicity, we will in the following assume that the costs are quadratic (L2), although linear (L1) costs in the firing rates could easily be accounted for as well. We note that autoencoder networks are sometimes assumed to reduce the dimensionality of the input (undercomplete case, $N < K$) and sometimes assumed to increase the dimensionality (overcomplete case, $N > K$). Our results apply to both cases.

The optimal set of firing rates for a given input signal can then be found by minimizing the loss function,

$$L = \frac{1}{2} \|\mathbf{x} - \mathbf{Dr}\|^2 + \frac{\mu}{2} \|\mathbf{r}\|^2, \tag{2}$$

with respect to the firing rates $\mathbf{r}$. Here, the first term is the error between the reconstructed input signal, $\hat{\mathbf{x}} = \mathbf{Dr}$, and the actual stimulus, $\mathbf{x}$, while the second term corresponds to the "cost" of the signal representation. The minimization can be carried out via gradient descent, resulting in the differential equation

$$\dot{\mathbf{r}} = -\frac{\partial L}{\partial \mathbf{r}} = -\mu\mathbf{r} + \mathbf{D}^\top \mathbf{x} - \mathbf{D}^\top \mathbf{Dr}. \tag{3}$$

This differential equation can be interpreted as a neural network with a 'leak', $-\mu\mathbf{r}$, feedforward connections, $\mathbf{F} = \mathbf{D}^T$, and recurrent connections, $\mathbf{\Omega} = \mathbf{D}^\top \mathbf{D}$. The derivation of neural networks from quadratic loss functions was first introduced by Hopfield [7, 8], and the link to the autoencoder was pointed out in [19]. Here, we have chosen a quadratic cost term which results in a linear differential equation. Depending on the precise nature of the cost term, one can also obtain non-linear differential equations, such as the Cowan-Wilson equations [19, 8]. Here, we will first focus on linear networks, in which case 'firing rates' can be both positive and negative. Further below, we will also show how our results can be generalized to networks with positive firing rates and to networks in which neurons spike.

In the case of arbitrarily small costs, the network can be understood as implementing predictive coding [17]. The reconstructed ("predicted") input signal, $\hat{\mathbf{x}} = \mathbf{Dr}$, is subtracted from the actual input signal, $\mathbf{x}$, see Fig. 1b. Predictive coding here enforces a cancellation or 'balance' between the feedforward and recurrent synaptic inputs. If we assume that the actual input acts excitatory, for instance, then the predicted input is mediated through recurrent lateral inhibition. Recent work has shown that this cancellation can be mediated by the detailed balance of currents in spiking networks [3, 1], a result we will return to later on.

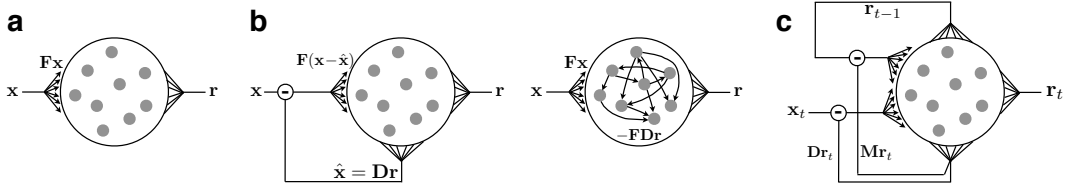

**Figure 1:** Autoencoders. (a) Feedforward network. The input signal $\mathbf{x}$ is multiplied with the feedforward weights $\mathbf{F}$. The network generates output firing rates $\mathbf{r}$. (b) Recurrent network. The left panel shows how the reconstructed input signal $\hat{\mathbf{x}} = \mathbf{Dr}$ is fed back and subtracted from the original input signal $\mathbf{x}$. The right panel shows that this subtraction can also be performed through recurrent connections $\mathbf{FD}$. For the optimal network, we set $\mathbf{F} = \mathbf{D}^{\top}$. (c) Recurrent network with delayed feedback. Here, the output firing rates are fed back with a delay. This delayed feedback acts as just another input signal, and is thereby re-used, thus generating short-term memory.

## 3   Unsupervised learning of the autoencoder with local learning rules

The transformation of the input signal, $\mathbf{x}$, into the output firing rate, $\mathbf{r}$, is largely governed by the decoder, $\mathbf{D}$, as can be seen in Eq. (3). When the inputs are drawn from a particular distribution, $p(\mathbf{x})$, such as the distribution of natural images or natural sounds, some decoders will lead to a smaller average loss and better performance. The average loss is given by

$$\langle L \rangle = \frac{1}{2}\big\langle\, \|\mathbf{x} - \mathbf{Dr}\|^2 + \mu\,\|\mathbf{r}\|^2 \,\big\rangle \tag{4}$$

where the angular brackets denote an average over many signal presentations. In practice, $\mathbf{x}$ will generally be centered and whitened. While it is straightforward to minimize this average loss with respect to the decoder, $\mathbf{D}$, biological networks face a different problem.[1] A general recurrent neural network is governed by the firing rate dynamics

$$\dot{\mathbf{r}} = -\mu\mathbf{r} + \mathbf{Fx} - \mathbf{\Omega r}, \tag{5}$$

and has therefore no access to the decoder, $\mathbf{D}$, but only to to its feedforward weights, $\mathbf{F}$, and its recurrent weights, $\mathbf{\Omega}$. Furthermore, any change in $\mathbf{F}$ and $\mathbf{\Omega}$ must solely relie on information that is locally available to each synapse.

We will assume that matrix $\mathbf{\Omega}$ is initially chosen such that the dynamical system is stable, in which case its equilibrium state is given by

$$\mathbf{Fx} = \mathbf{\Omega r} + \mu\mathbf{r}. \tag{6}$$

If the dynamics of the input signal $\mathbf{x}$ are slow compared to the firing rate dynamics of the autoencoder, the network will generally operate close to equilibrium. We will assume that this is the case, and show that this assumption helps us to bridge from these firing rate networks to spiking networks further below.

A priori, it is not clear how to change the feedforward weights, $\mathbf{F}$, or the recurrent weights, $\mathbf{\Omega}$, since neither appears in the average loss function, Eq. (4). We might be inclined to solve Eq. (6) for $\mathbf{r}$ and plug the result into Eq. (4). However, we then have to operate on matrix inverses, the resulting gradients imply heavily non-local synaptic operations, and we would still need to somehow eliminate the decoder, $\mathbf{D}$, from the picture.

Here, we follow a different approach. We note that the optimal target network in the previous section implements a form of predictive coding. We therefore suggest a two-step approach to the learning problem. First, we fix the feedforward weights and we set up learning rules for the recurrent weights such that the network moves into a regime where the inputs, $\mathbf{Fx}$, are predicted or 'balanced' by the recurrent weights, $\mathbf{\Omega r}$, see Fig. 1b. In this case, $\mathbf{\Omega} = \mathbf{FD}$, and this will be our first target for learning. Second, once $\mathbf{\Omega}$ is learnt, we change the feedforward weights $\mathbf{F}$ to decrease the average loss even further. We then return to step 1 and iterate.

Since $\mathbf{F}$ is assumed constant in step 1, we can reach the target $\mathbf{\Omega} = \mathbf{FD}$ by investigating how the decoder $\mathbf{D}$ needs to change. The respective learning equation for $\mathbf{D}$ can then be translated into a learning equation for $\mathbf{\Omega}$, which will directly link the learning of $\mathbf{\Omega}$ to the minimization of the loss function, Eq. (4). One thing to keep in mind, however, is that any change in $\mathbf{\Omega}$ will cause a compensatory change in $\mathbf{r}$ such that Eq. (6) remains fulfilled. These changes are related through the equation

$$\dot{\mathbf{\Omega}}\mathbf{r} + (\mathbf{\Omega} + \mu\mathbf{I})\dot{\mathbf{r}} = 0 \tag{7}$$

which is obtained by taking the derivative of Eq. (6) and remembering that $\mathbf{x}$ changes on much slower time scales, and can therefore be considered a constant. In consequence, we have to consider the combined change of the recurrent weights, $\mathbf{\Omega}$, and the equilibrium firing rate, $\mathbf{r}$, in order to reduce the average loss.

Let us assume a small change of $\mathbf{D}$ in the direction $\Delta\mathbf{D} = \epsilon\mathbf{xr}^\top$, which is equivalent to simply decreasing $\mathbf{x}$ in the first term of Eq. (4). Such a small change can be translated into the following learning rule for $\mathbf{D}$,

$$\dot{\mathbf{D}} = \epsilon(\mathbf{xr}^\top - \alpha\mathbf{D}), \tag{8}$$

where $\epsilon$ is sufficiently small to make the learning slow compared to the dynamics of the input signals $\mathbf{x} = \mathbf{x}(t)$. The 'weight decay' term, $-\alpha\mathbf{D}$, acts as a soft normalization or regularizer on $\mathbf{D}$. In turn, to have the recurrent weights $\mathbf{\Omega}$ move towards $\mathbf{FD}$, we multiply with $\mathbf{F}$ from the left to obtain the learning rule[2]

$$\dot{\mathbf{\Omega}} = \epsilon(\mathbf{Fxr}^\top - \alpha\mathbf{\Omega}). \tag{9}$$

Importantly, this learning rule is completely local: it only rests on information that is available to each synapse, namely the presynaptic firing rates, $\mathbf{r}$, and the postsynaptic input signal, $\mathbf{Fx}$.

Finally, we show that the 'unnormalized' learning rule decreases the loss function. As noted above, any change of $\mathbf{\Omega}$ causes a change in the equilibrium firing rate, see Eq. (7). By plugging the unnormalized learning rule for $\mathbf{\Omega}$, namely $\epsilon\mathbf{Fxr}^\top$, into Eq. (7), and by remembering that $\mathbf{Fx} = \mathbf{\Omega r} + \mu\mathbf{r}$, we obtain

$$\dot{\mathbf{r}} = -\epsilon\|\mathbf{r}\|^2\mathbf{r}. \tag{10}$$

So, to first order, the firing rates decay in the direction of $\mathbf{r}$. In turn, the temporal derivative of the loss function,

$$\frac{d\langle L \rangle}{dt} = \left\langle (-\dot{\mathbf{D}}\mathbf{r} - \mathbf{D}\dot{\mathbf{r}})^\top(\mathbf{x} - \mathbf{Dr}) + \mu\dot{\mathbf{r}}^T\mathbf{r} \right\rangle \tag{11}$$

$$= \left\langle -\epsilon\|\mathbf{r}\|^2(\mathbf{x} - \mathbf{Dr})^\top(\mathbf{x} - \mathbf{Dr}) - \mu\epsilon\|\mathbf{r}\|^4 \right\rangle, \tag{12}$$

is always negative so that the unnormalized learning rule for $\mathbf{\Omega}$ decreases the error. We then subtract the term $-\epsilon\alpha\mathbf{\Omega}$ (thus reducing the norm of the matrix but not changing the direction) as a 'soft normalisation' to prevent it from going to infinity. Note that the argument here rests on the parallelism of the learning of $\mathbf{D}$ and $\mathbf{\Omega}$. The decoder, $\mathbf{D}$, however, is merely a hypothetical quantity that does not have a physical counterpart in the network.

In step 2, we assume that the recurrent weights have reached their target, $\mathbf{\Omega} = \mathbf{FD}$, and we learn the feedforward weights. For that we notice that in the absolute minimum, as shown in the previous section, the feedforward weights become $\mathbf{F} = \mathbf{D}^\top$. Hence, the target for the feedforward weights should be the transpose of the decoder. Over long time intervals, the expected decoder is simply $\mathbf{D} = \langle\mathbf{xr}^\top\rangle/\alpha$, since that is the fixed point of the decoder learning rule, Eq. (8). Hence, we suggest to learn the feedforward weights on a yet slower time scale $\beta \ll \epsilon$, according to

$$\dot{\mathbf{F}} = \beta(\mathbf{rx}^\top - \lambda\mathbf{F}), \tag{13}$$

where $\lambda\mathbf{F}$ is once more a soft normalization factor. The fixed point of the learning rule is then $\mathbf{F} = \mathbf{D}^\top$. We emphasize that this learning rule is also local, based solely on the presynaptic input signal and postsynaptic firing rates.

In summary, we note that the autoencoder operates on four separate time scales. On a very fast, almost instantaneous time scale, the firing rates run into equilibrium for a given input signal, Eq. (6). On a slower time scale, the input signal, $\mathbf{x}$, changes. On a yet slower time scale, the recurrent weights, $\mathbf{\Omega}$, are learnt, and their learning therefore uses many input signal values. On the final and slowest time scale, the feedforward weights, $\mathbf{F}$, are optimized.

## 4 Unsupervised learning for non-whitened inputs

Algorithms for efficient coding are generally applied to whitened and centered data (see e.g. [2, 16]). Indeed, if the data are not centered, the read-out of the neurons will concentrate in the direction of the mean input signal in order to represent it, even though the mean may not carry any relevant information about the actual, time-varying signal. If the data are not whitened, the choice of the decoder will be dominated by second-order statistical dependencies, at the cost of representing higher-order dependencies. The latter are often more interesting to represent, as shown by applications of efficient or sparse coding algorithms to the visual system [20].

While whitening and centering are therefore common pre-processing steps, we note that, with a simple correction, our autoencoder network can take care of the pre-processing steps autonomously. This extra step will be crucial later on, when we feed the time-delayed (and non-whitened) network activity back into the network. The main idea is simple: we suggest to use a cost function that is invariant under affine transformations *and* equals the cost function we have been using until now in case of centered and whitened data. To do so, we introduce the short-hands $\mathbf{x}_c = \mathbf{x} - \langle \mathbf{x} \rangle$ and $\mathbf{r}_c = \mathbf{r} - \langle \mathbf{r} \rangle$ for the centered input and the centered firing rates, and we write $\mathbf{C} = \mathrm{cov}(\mathbf{x}, \mathbf{x})$ for the covariance matrix of the input signal. The corrected loss function is then,

$$ L = \frac{1}{2}(\mathbf{x}_c - \mathbf{D}\mathbf{r}_c)^\top \mathbf{C}^{-1}(\mathbf{x}_c - \mathbf{D}\mathbf{r}_c) + \frac{\mu}{2}\|\mathbf{r}\|^2 . \tag{14} $$

The loss function reduces to Eq. (2) if the data are centered and if $\mathbf{C} = \mathbf{I}$. Furthermore, the value of the loss function remains constant if we apply any affine transformation $\mathbf{x} \rightarrow \mathbf{A}\mathbf{x} + \mathbf{b}$.[3] In turn, we can interpret the loss function as the likelihood function of a Gaussian.

From hereon, we can follow through exactly the same derivations as in the previous sections. We first notice that the *optimal* firing rate dynamics becomes

$$ \mathbf{V} = \mathbf{D}^\top \mathbf{C}^{-1}\mathbf{x} - \mathbf{D}^\top \mathbf{C}^{-1}\mathbf{D}\mathbf{r} - \mu\mathbf{r} \tag{15} $$
$$ \dot{\mathbf{r}} = \mathbf{V} - \langle \mathbf{V} \rangle \tag{16} $$

where $\mathbf{V}$ is a placeholder for the overall input. The dynamics differ in two ways from those in Eq. (3). First, the dynamics now require the subtraction of the averaged input, $\langle \mathbf{V} \rangle$. Biophysically, this subtraction could correspond to a slower intracellular process, such as adaptation through hyperpolarization. Second, the optimal feedforward weights are now $\mathbf{F} = \mathbf{D}^\top \mathbf{C}^{-1}$, and the optimal recurrent weights become $\mathbf{\Omega} = \mathbf{D}^\top \mathbf{C}^{-1}\mathbf{D}\mathbf{r}$.

The derivation of the learning rules follows the outline of the previous section. Initially, the network starts with some random connectivity, and obeys the dynamical equations,

$$ \mathbf{V} = \mathbf{F}\mathbf{x} - \mathbf{\Omega}\mathbf{r} - \mu\mathbf{r} \tag{17} $$
$$ \dot{\mathbf{r}} = \mathbf{V} - \langle \mathbf{V} \rangle . \tag{18} $$

We then apply the following modified learning rules for $\mathbf{D}$ and $\mathbf{\Omega}$,

$$ \dot{\mathbf{D}} = \epsilon\big(\mathbf{x}\mathbf{r}^\top - \langle \mathbf{x} \rangle \langle \mathbf{r} \rangle^\top - \alpha\mathbf{D}\big) \tag{19} $$
$$ \dot{\mathbf{\Omega}} = \epsilon\big(\mathbf{F}\mathbf{x}\mathbf{r}^\top - \langle \mathbf{F}\mathbf{x} \rangle \langle \mathbf{r} \rangle^\top - \alpha\mathbf{\Omega}\big) . \tag{20} $$

We note that in both cases, the learning remains local. However, similar to the rate dynamics, the dynamics of learning now requires a slower synaptic process that computes the averaged signal inputs and presynaptic firing rates. Synapses are well-known to operate on a large range of time scales (e.g., [5]), so that such slower processes are in broad agreement with physiology.

The target for learning the feedforward weights becomes $\mathbf{F} \rightarrow \mathbf{D}^\top \mathbf{C}^{-1}$. The matrix inverse can be eliminated by noticing that the differential equation $\dot{\mathbf{F}} = \epsilon(-\mathbf{F}\mathbf{C} + \mathbf{D}^\top)$ has the required target as its fixed point. The covariance matrix $\mathbf{C}$ can be estimated by averaging over $\mathbf{x}_c\mathbf{x}_c^\top$, and the decoder $\mathbf{D}^\top$ can be estimated by averaging over $\mathbf{x}_c\mathbf{r}_c^\top$, just as in the previous section, or as follows from Eq. (19). Hence, the learning of the feedforward weights becomes

$$ \dot{\mathbf{F}} = \beta\Big((\mathbf{r} - \mathbf{F}\mathbf{x})\mathbf{x}^\top - \langle \mathbf{r} - \mathbf{F}\mathbf{x} \rangle \langle \mathbf{x}^\top \rangle - \alpha\mathbf{F}\Big) . \tag{21} $$

As for the recurrent weights, the learning rests on local information, but requires a slower time scale that computes the mean input signal and presynaptic firing rates.

# 5 The autoencoder with memory

We are finally in a position to tackle the problem we started out with, how to build a recurrent network that efficiently represents not just its present input, but also its past inputs. The objective function used so far, however, completely neglects the input history: even if the dimensionality of the input is much smaller than the number of neurons available to code it, the network will not try to use the extra 'space' available to remember the input history.

## 5.1 An objective function for short-term memory

Ideally, we would want to be able to read out both the present input and the past inputs, such that $\mathbf{x}_{t-n} \approx \mathbf{D}_n \mathbf{r}_t$, where $n$ is an elementary time step, and $\mathbf{D}_n$ are appropriately chosen readouts. We will in the following assume that there is a matrix $\mathbf{M}$ such that $\mathbf{D}_n \mathbf{M} = \mathbf{D}_{n+1}$ for all $n$. In other words, the input history should be accessible via $\hat{\mathbf{x}}_{t-n} = \mathbf{D}_n \mathbf{r} = \mathbf{D}_0 \mathbf{M}^n \mathbf{r}_t$. Then the cost function we would like to minimize is a straightforward generalization of Eq. (2),

$$L = \frac{1}{2} \sum_{n=0} \gamma^n \|\mathbf{x}_{t-n} - \mathbf{D}\mathbf{M}^n \mathbf{r}_t\|^2 + \frac{\mu}{2} \|\mathbf{r}_t\|^2. \tag{22}$$

where we have set $\mathbf{D} = \mathbf{D}_0$. We tacitly assume that $\mathbf{x}$ and $\mathbf{r}$ are centered and that the L2 norm is defined with respect to the input signal covariance matrix $\mathbf{C}$, so that we can work in the full generality of Eq. (14) without keeping the additional notational baggage.

Unfortunately, the direct minimization of this objective is impossible, since the network has no access to the past inputs $\mathbf{x}_{t-n}$ for $n \geq 1$. Rather, information about past inputs will have to be retrieved from the network activity itself. We can enforce that by replacing the past input signal at time $t$, with its estimate in the previous time step, which we will denote by a prime. In other words, instead of asking that $\mathbf{x}_{t-n} \approx \hat{\mathbf{x}}_{t-n}$, we ask that $\hat{\mathbf{x}}'_{(t-1)-(n-1)} \approx \hat{\mathbf{x}}_{t-n}$, so that the estimates of the input (and its history) are properly propagated through the network. Given the iterative character of the respective errors, $\|\hat{\mathbf{x}}'_{(t-1)-(n-1)} - \hat{\mathbf{x}}_{t-n}\| = \|\mathbf{D}\mathbf{M}^{n-1}(\mathbf{r}_{t-1} - \mathbf{M}\mathbf{r}_t)\|$, we can define a loss function for one time step only,

$$L = \frac{1}{2} \|\mathbf{x}_t - \mathbf{D}\mathbf{r}_t\|^2 + \frac{\gamma}{2} \|\mathbf{r}_{t-1} - \mathbf{M}\mathbf{r}_t\|^2 + \frac{\mu}{2} \|\mathbf{r}_t\|^2. \tag{23}$$

Here, the first term enforces that the instantaneous input signal is properly encoded, while the second term ensures that the network is remembering past information. The last term is a cost term that makes the system more stable and efficient.

Note that a network which minimizes this loss function is maximizing its information content, even if the number of neurons, $N$, far exceeds the input dimension $K$, so that $N \gg K$. As becomes clear from inspecting the loss function, the network is trying to code an $N + K$ dimensional signal with only $N$ neurons. Consequently, just as in the undercomplete autoencoder, all of its information capacity will be used.

## 5.2 Dynamics and learning

Conceptually, the loss function in Eq. (23) is identical to Eq. (2), or rather, to Eq. (14), if we keep full generality. We only need to vertically stack the feedforward input and the delayed recurrent input into a single high-dimensional vector $\mathbf{x}' = (\mathbf{x}_t \; ; \; \gamma \mathbf{r}_{t-1})$. Similarly, we can horizontally combine the decoder $\mathbf{D}$ and the 'time travel' matrix $\mathbf{M}$ into a single decoder matrix $\mathbf{D}' = (\mathbf{D} \quad \gamma \mathbf{M})$. The above loss function then reduces to

$$L = \|\mathbf{x}'_t - \mathbf{D}'\mathbf{r}_t\|^2 + \mu \|\mathbf{r}_t\|^2, \tag{24}$$

and all of our derivations, including the learning rules, can be directly applied to this system. Note that the 'input' to the network now combines the actual input signal, $\mathbf{x}_t$, and the delayed recurrent input, $\mathbf{r}_{t-1}$. Consequently, this extended input is neither white nor centered, and we will need to work with the generalized dynamics and generalized learning rules derived in the previous section.

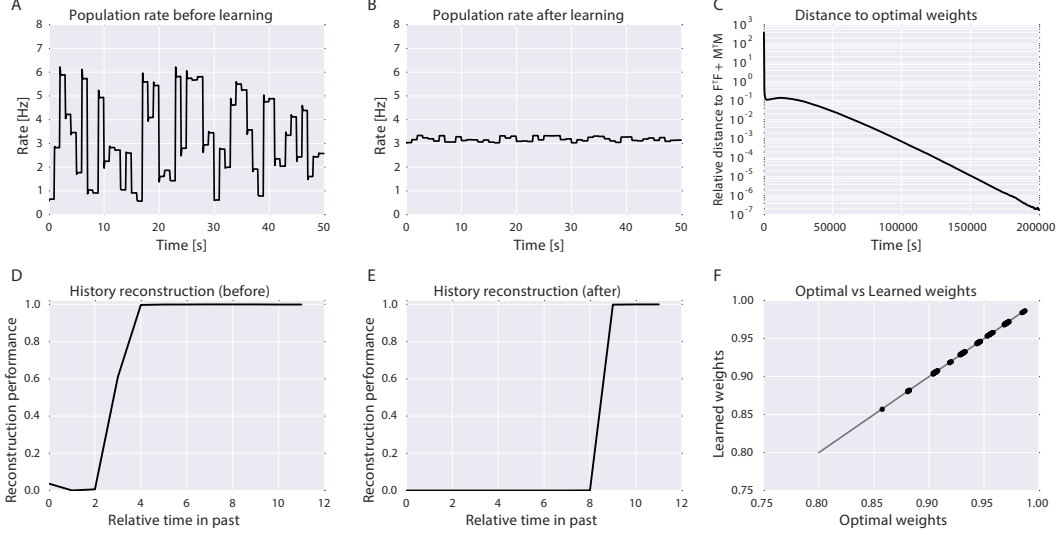

**Figure 2:** Emergence of working memory in a network of 10 neurons with random initial connectivity. **(A)** Rates of all neurons for the first 50 inputs at the beginning of learning. **(B)** Same as (A), but after learning. **(C)** Distance of fast recurrent weights to optimal configuration, $\mathbf{F}^\top\mathbf{F} + \mathbf{M}^\top\mathbf{M}$, relative to L2-norm of optimal weights. **(D)** Squared error of optimal linear reconstruction of inputs at time $t - k$ from rates at time $t$, relative to variance of the input before learning; for $k \in [0, \dots, 20]$. **(E)** Same as (D) but after learning. **(F)** Scatter plot of fast recurrent weights after learning against optimal configuration, $\mathbf{F}^\top\mathbf{F} + \mathbf{M}^\top\mathbf{M}$.

The network dynamics will initially follow the differential equation [4]

$$\mathbf{V} = \mathbf{F}\mathbf{x}_t + \mathbf{\Omega}^d\mathbf{r}_{t-1} - \mathbf{\Omega}^f\mathbf{r}_t - \mu\mathbf{r}_t \tag{25}$$

$$\dot{\mathbf{r}} = \mathbf{V} - \langle\mathbf{V}\rangle. \tag{26}$$

Compared to our previous network, we now have effectively three inputs into the network: the feedforward inputs with weight $\mathbf{F}$, a delayed recurrent input with weight $\mathbf{\Omega}^d$ and a fast recurrent input with weight $\mathbf{\Omega}^f$, see Fig. 1c. The optimal connectivities can be derived from the loss function and are (see also Fig. 1c)

$$\mathbf{F} = \mathbf{D}^\top \tag{27}$$

$$\mathbf{\Omega}^d = \mathbf{M}^\top \tag{28}$$

$$\mathbf{\Omega}^f = \mathbf{D}^\top\mathbf{D} + \mathbf{M}^\top\mathbf{M}. \tag{29}$$

Consequently, there are also three learning rules: one for the fast recurrent weights, which follows Eq. (20), one for the feedforward weights, which follows Eq. (21), and one for the delayed recurrent weights, which also follows Eq. (21). In summary,

$$\dot{\mathbf{\Omega}}^f = \epsilon(\mathbf{F}\mathbf{x}_t\mathbf{r}_t^\top - \langle\mathbf{F}\mathbf{x}_t\rangle\langle\mathbf{r}\rangle^\top - \alpha\mathbf{\Omega}) \tag{30}$$

$$\dot{\mathbf{F}} = \beta\big((\mathbf{r}_t - \mathbf{F}\mathbf{x}_t)\mathbf{x}_t^\top - \langle\mathbf{r}_t - \mathbf{F}\mathbf{x}_t\rangle\langle\mathbf{x}_t^\top\rangle - \alpha\mathbf{F}\big). \tag{31}$$

$$\dot{\mathbf{\Omega}}^d = \beta\big((\mathbf{r}_t - \mathbf{\Omega}^d\mathbf{r}_{t-1})\mathbf{r}_{t-1}^\top - \langle\mathbf{r}_t - \mathbf{\Omega}^d\mathbf{r}_{t-1}\rangle\langle\mathbf{r}_{t-1}^\top\rangle - \alpha\mathbf{\Omega}^d\big). \tag{32}$$

We note that the learning of the slow connections does not strictly minimize the expected loss in every time step, due to potential non-stationarities in the distribution of firing rates throughout the course of learning. In practice, we therefore find that the improvement in memory performance is often dominated by the learning of the fast connectivity (see example below).

## 6 Simulations

We simulated a firing rate network of ten neurons that learn to remember a one-dimensional, temporally uncorrelated white noise stimulus (Fig. 2). Firing rates were constrained to be positive. We initialized all feedforward weights to one, whereas the matrices $\mathbf{\Omega}^f$ and $\mathbf{\Omega}^d$ were initialised by drawing numbers from centered Gaussian distributions with variance 1 and 0.2 respectively. All matrices were then divided by $N^2 = 100$. At the onset, the network has some memory, similar to random networks based on reservoir computing. However, the recurrent inputs are generally not cancelling out the feedforward inputs. The effect of such imprecise balance are initially high firing rates and poor coding properties (Fig. 2A,D). At the end of learning, neurons are firing less, and the coding properties are close to the information-theoretic limit (10 time steps), see Fig. 2B,E. We note that, although the signal input was white noise for simplicity, the total input into the network (i.e., including the delayed firing rates) is neither white nor zero-mean, due to the positivity constraint on the firing rates. The network converges to the derived connectivity (Fig. 2C,F); we note, however, that the bulk of the improvements is due to the learning of the fast connections.

## 7 Towards learning in spiking recurrent networks

While we have shown how a recurrent network can learn to efficiently represent an input and its history using only *local* learning rules, our network is still far from being biologically realistic. A quite obvious discrepancy with biological networks is that the neurons are not spiking, but rather emit 'firing rates' that can be both positive and negative. How can we make the connection to spiking networks? Standard solutions have bridged from rate to spiking networks using mean-field approaches [18]. However, more recent work has shown that there is a direct link from the types of loss functions considered in this paper to *balanced* spiking networks.

Recently, Hu et al. pointed out that the minimization of Eq. (2) can be done by a network of neurons that fires both positive and negative spikes [9], and then argued that these networks can be translated into real spiking networks. A similar, but more direct approach was introduced in [3, 1] who suggested to minimize the loss function, Eq. (2), under the constraint that $\mathbf{r} \geq 0$. The resulting networks consist of recurrently connected integrate-and-fire neurons that balance their feedforward and recurrent inputs [3, 1, 4]. Importantly, Eq. (2) remains a convex function of $\mathbf{r}$, and Eq. (3) still applies (except that $\mathbf{r}$ cannot become negative).

The precise match between the spiking network implementation and the firing rate minimization [1] opens up the possibility to apply our learning rules to the spiking networks. We note, though, that this holds only strictly in the regime where the spiking networks are balanced. (For unbalanced networks, there is no direct link to the firing rate formalism.) If the initial network is not balanced, we need to first learn how to bring it into the balanced state. For white-noise Gaussian inputs, [4] showed how this can be done. For more general inputs, this problem will have to be solved in the future.

## 8 Discussion

In summary, we have shown how a recurrent neural network can learn to efficiently represent both its present and past inputs. A key insight has been the link between balancing of feedforward and recurrent inputs and the minimization of the cost function. If neurons can compensate both external feedforward and delayed recurrent excitation with lateral inhibition, then, to some extent, they must be coding the temporal trajectory of the stimulus. Indeed, in order to be able to compensate an input, the network must be coding it at some level. Furthermore, if synapses are linear, then so must be the decoder.

We have shown that this 'balance' can be learnt through local synaptic plasticity of the lateral connections, based only on the presynaptic input signals and postsynaptic firing rates of the neurons. Performance can then be further improved by learning the feedforward connections (as well as the 'time travel' matrix) which thereby take the input statistics into account. In our network simulations, these connections only played a minor role in the overall improvements. Since the learning rules for the time-travel matrix do not strictly minimize the expected loss (see above), there may still be room for future improvements.

## Footnotes

*current address: Centre for Integrative Neuroscience, University of Tübingen, Germany

[1]Note that minimization of the average loss with respect to $\mathbf{D}$ requires either a hard or a soft normalization constraint on $\mathbf{D}$.

[2]Note that the fixed point of the decoder learning rule is $\mathbf{D} = \langle\mathbf{xr}^\top\rangle/\alpha$. Hence, the fixed point of the recurrent learning is $\mathbf{\Omega} = \mathbf{FD}$.

[3]Under an affine transformation, $\mathbf{y} = \mathbf{A}\mathbf{x} + \mathbf{b}$ and $\hat{\mathbf{y}} = \mathbf{A}\hat{\mathbf{x}} + \mathbf{b}$, we obtain: $(\mathbf{y} - \hat{\mathbf{y}})^\top \mathrm{cov}(\mathbf{y}, \mathbf{y})^{-1}(\mathbf{y} - \hat{\mathbf{y}}) = (\mathbf{A}\mathbf{x} - \mathbf{A}\hat{\mathbf{x}})^\top \mathrm{cov}(\mathbf{A}\mathbf{x}, \mathbf{A}\mathbf{x})^{-1}(\mathbf{A}\mathbf{x} - \mathbf{A}\hat{\mathbf{x}}) = (\mathbf{x} - \hat{\mathbf{x}})^\top \mathrm{cov}(\mathbf{x}, \mathbf{x})^{-1}(\mathbf{x} - \hat{\mathbf{x}}).$

[4]We are now dealing with a delay-differential equation, which may be obscured by our notation. In practice, the term $\mathbf{r}_{t-1}$ would be replaced by a term of the form $\mathbf{r}(r - \tau)$, where $\tau$ is the actual value of the 'time step'.

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
