[Supplementary Material]

# Supplementary Material
## Unsupervised learning of an efficient short-term memory network

Pietro Vertechi*, Wieland Brendel* and Christian K. Machens

We assume zero-mean inputs and zero temporal input correlations throughout this supplementary material. W.l.o.g. the covariance of the input $\mathbf{x}_t$ has full rank (otherwise there exists an equivalent dynamical system with full rank). Equation references without prefix refer to the main text.

## 1 FIXED POINTS OF AUTOENCODER

The optimal synaptic connectivity that minimizes the reconstruction error (14) is given by (cp. with eq. (15))

$$\mathbf{F}^*\mathbf{C}_x = \mathbf{D}^{*\top}, \tag{S.1}$$

$$\mathbf{\Omega}^* = \mathbf{D}^{*\top}\mathbf{C}_x^{-1}\mathbf{D}^*, \tag{S.2}$$

$$\mathbf{D}^* \propto \sqrt{\mathbf{C}_x}\,\mathbf{U}, \tag{S.3}$$

where $\mathbf{U} \in \mathbb{R}^{K \times N}$ is orthonormal. Here the last property ensures that the signal is actively whitened before being encoded along the orthogonal axis.

Do the synaptic plasticity rules (20, 21) lead the network into this optimal topology? Proving convergence is difficult for recurrent networks with plastic synapses (but simulations converge robustly and reliably). We can, however, determine the fixed points of the synaptic dynamics and analyse their stability. To determine the fixed points, we note from (19-21) that the synaptic weights remain stationary whenever the following conditions are met,

$$\alpha\mathbf{D} = \left\langle \mathbf{x}\mathbf{r}^\top \right\rangle,$$

$$\alpha\mathbf{F} = \mathbf{D}^\top\mathbf{C}_x^{-1}, \tag{S.4}$$

$$\alpha\mathbf{\Omega} = \frac{1}{\alpha}\mathbf{F}\mathbf{D} = \mathbf{D}^\top\mathbf{C}_x^{-1}\mathbf{D}. \tag{S.5}$$

As is directly evident from these equations, the stationary synaptic weights fulfil the first two optimality conditions (S.1) and (S.2). But are the columns of $\mathbf{D}$ orthogonal (up to whitening)?

To solve for the fixed points of $\mathbf{D}$, we replace the neural activities $\mathbf{r}$ with their equilibrium state determined from the balance condition (17),

$$\mathbf{r} = (\mathbf{\Omega} + \mu\mathbf{I})^{-1}\mathbf{F}\mathbf{x}.$$

Then the fixed point of the decoder is given by,

$$\alpha\mathbf{D} = \left\langle \mathbf{x}\mathbf{x}^\top \right\rangle \mathbf{F}^\top \left(\mathbf{\Omega}^\top + \mu\mathbf{I}\right)^{-1},$$

$$= \mathbf{C}_x\mathbf{F}^\top \left(\mathbf{\Omega}^\top + \mu\mathbf{I}\right)^{-1},$$

in which we can replace $\mathbf{\Omega}$ and $\mathbf{F}$ with their converged states (S.4) and (S.5),

$$= \mathbf{D}\left(\mathbf{D}^\top\mathbf{C}_x^{-1}\mathbf{D} + \alpha\mu\mathbf{I}\right)^{-1}.$$

To get rid of the matrix inverse, we multiply with $\left( \mathbf{D}^\top \mathbf{C}_x^{-1} \mathbf{D} + \alpha\mu\mathbf{I} \right)$ from the right and sort by products of $\mathbf{D}$,

$$\alpha\mathbf{DD}^\top \mathbf{C}_x^{-1}\mathbf{D} = (1 - \alpha\mu)\,\mathbf{D}. \tag{S.6}$$

If $\mathbf{D}$ is of full rank, then (S.6) simplifies: If the dimension $K$ of the signal is less then the number $N$ of neurons, then multiply (S.6) with $\mathbf{D}^+$ from the right,

$$\mathbf{DD}^\top \mathbf{C}_x^{-1} = (\alpha^{-1} - \mu)\mathbf{I}.$$

If $K \geq N$, multiply from the left,

$$\mathbf{DC}_x^{-1}\mathbf{D}^\top = (\alpha^{-1} - \mu)\mathbf{I}.$$

In both cases the solutions satisfy (S.3). As a result, if $\mathbf{D}$ is full rank then all fixed points, to which the network could converge, fulfil the optimality conditions (S.1)-(S.3). If $\mathbf{D}$ is not full rank, then all corresponding fixed points are unstable, as one can see from the dynamics of small perturbations of the fast connectivity. We work out the full proof in the more demanding setting of a working memory network.

## 2 STABLE FIXED POINTS OF WORKING MEMORY

For the autoencoder, the best network topology is fully determined by the optimal decoder $\mathbf{D}$. The working memory depends additionally on the delayed projections $\mathbf{M}$, which makes it harder to determine the optimal network configurations.

As a proxy for optimality we rely on the amount of information stored in the network state. For temporally independent inputs $\mathbf{x}_t$ the maximum effective number of steps the network can store is equal to the number of neurons. In other words, the network is optimal if the covariance matrix $\mathbf{C}_r$ is proportional to the identity. In this section, we prove that all stable fixed points of the synaptic weights fulfil this property.

As for the autoencoder, the effective fixed points for the signal and history decoder are given by $\alpha\mathbf{D} = \left\langle \mathbf{x}_t \mathbf{r}_t^\top \right\rangle$ and $\alpha\mathbf{M} = \left\langle \mathbf{r}_{t-1}\mathbf{r}_t^\top \right\rangle$. The fixed points of the synaptic weights can be directly determined from the learning rules (30)-(32),

$$\alpha\mathbf{FC}_x = \left\langle \mathbf{r}_t\mathbf{x}_t \right\rangle = \alpha\mathbf{D}^\top, \tag{S.7}$$

$$\boldsymbol{\Omega}^f = \mathbf{FD} + \boldsymbol{\Omega}^d\mathbf{M},$$

$$\alpha\boldsymbol{\Omega}^d\mathbf{C}_r = \left\langle \mathbf{r}_{t-1}\mathbf{r}_t^\top \right\rangle = \alpha\mathbf{M}^\top. \tag{S.8}$$

Note that $\mathbf{C}_r$ is not necessarily full rank. From the rate dynamics (25) we can determine the equilibrium of the network state at each time $t$,

$$\mathbf{r}_t = \mathbf{Q}\left( \mathbf{Fx}_t + \boldsymbol{\Omega}^d\mathbf{r}_{t-1} \right). \tag{S.9}$$

where we defined $\mathbf{Q} = \left( \boldsymbol{\Omega}^f + \mu\mathbf{I} \right)^{-1}$ for notational convenience. Plugging (S.9) into the fixed points (S.7) and (S.8) yields two important properties,

$$\mathbf{QFC}_x = \alpha\mathbf{FC}_x, \tag{S.10}$$

$$\mathbf{Q}\boldsymbol{\Omega}^d\mathbf{C}_r = \alpha\boldsymbol{\Omega}^d\mathbf{C}_r. \tag{S.11}$$

These two properties are important to solve explicitly for the rate covariance matrix $\mathbf{C}_r$. To this end, we first note that the network state $\mathbf{r}_t$ (S.9) at time $t$ is the result of the whole history of inputs,

$$\mathbf{r}_t = \sum_{k=0}^{\infty} (\mathbf{Q}\boldsymbol{\Omega}^d)^k \mathbf{QFx}_{t-k},$$

and so

$$\mathbf{C}_r = \left\langle \mathbf{r}_t \mathbf{r}_t^\top \right\rangle,$$
$$= \sum_{k=0}^{\infty} (\mathbf{Q}\mathbf{\Omega}^d)^k \mathbf{Q}\mathbf{F}\mathbf{C}_x \mathbf{F}^\top \mathbf{Q}^\top (\mathbf{Q}\mathbf{\Omega}^d)^{k\top},$$
$$= \mathbf{Q}\mathbf{F}\mathbf{C}_x \mathbf{F}^\top \mathbf{Q}^\top + \sum_{k=1}^{\infty} (\mathbf{Q}\mathbf{\Omega}^d)^k \mathbf{Q}\mathbf{F}\mathbf{C}_x \mathbf{F}^\top \mathbf{Q}^\top (\mathbf{Q}\mathbf{\Omega}^d)^{k\top},$$
$$= \mathbf{Q}\mathbf{F}\mathbf{C}_x \mathbf{F}^\top \mathbf{Q}^\top + \mathbf{Q}\mathbf{\Omega}^d \left( \sum_{k=0}^{\infty} (\mathbf{Q}\mathbf{\Omega}^d)^k \mathbf{Q}\mathbf{F}\mathbf{C}_x \mathbf{F}^\top \mathbf{Q}^\top (\mathbf{Q}\mathbf{\Omega}^d)^{k\top} \right) \mathbf{\Omega}^{d\top} \mathbf{Q}^\top,$$
$$= \mathbf{Q}\mathbf{F}\mathbf{C}_x \mathbf{F}^\top \mathbf{Q}^\top + \mathbf{Q}\mathbf{\Omega}^d \mathbf{C}_r \mathbf{\Omega}^{d\top} \mathbf{Q}^\top.$$

This expression can be substantially simplified by using the properties (S.10) and (S.11),

$$= \alpha^2 \left( \mathbf{F}\mathbf{C}_x \mathbf{F}^\top + \mathbf{\Omega}^d \mathbf{C}_r \mathbf{\Omega}^{d\top} \right),$$
$$= \alpha^2 \left( \mathbf{F}\mathbf{D} + \mathbf{\Omega}^d \mathbf{M} \right),$$
$$= \alpha^2 \mathbf{\Omega}^f.$$

In other words, the fast synaptic weights directly reflect the pairwise rate covariance! This special property, together with (S.10) and (S.11), yields an important property of $\mathbf{C}_r$,

$$\mathbf{Q}\mathbf{C}_r = \alpha^2 \mathbf{Q}(\mathbf{F}\mathbf{D} + \mathbf{\Omega}^d \mathbf{M}) = \alpha \mathbf{C}_r,$$

and equally $\mathbf{Q}^{-1}\mathbf{C}_r = \alpha^{-1}\mathbf{C}_r$. This last property is the missing piece to prove that $\mathbf{C}_r$ is proportional to a projection matrix,

$$\mathbf{C}_r^2 = \alpha^2 \mathbf{\Omega}^f \mathbf{C}_r,$$
$$= \alpha^2 (\mathbf{\Omega}^f + \mu\mathbf{I} - \mu\mathbf{I})\mathbf{C}_r,$$
$$= \alpha^2 (\mathbf{Q}^{-1} - \mu\mathbf{I})\mathbf{C}_r,$$
$$= \alpha(1 - \alpha\mu)\mathbf{C}_r.$$

If $\mathbf{C}_r$ is full rank, then

$$\mathbf{C}_r \propto \mathbf{I}. \tag{S.12}$$

But is the converged network full rank? If $\mathbf{C}_r$ is not full rank, then all associated fixed points are unstable. To prove this, we need to show that a small perturbation of the slow connections $\mathbf{\Omega}^d$ can lead to further changes $\dot{\mathbf{\Omega}}^d$ that are oriented away from the fixed point.

For notation, let $\mathbf{V}$ be the matrix of eigenvectors of $\mathbf{C}_r$ with non-zero eigenvalues, and let $\mathbf{U}$ be its orthogonal complement (i.e. $\mathbf{U}^\top \mathbf{V} = \mathbf{0}$). Let $\mathbf{v} \in \text{span}\,\mathbf{V}$ be in the subspace spanned by the firing rates, and $\mathbf{u} \in \text{span}\,\mathbf{U}$ in its complement. We want to prove that a small perturbation $\epsilon \mathbf{u}\mathbf{v}^\top$ makes the slow weights grow in the direction of that perturbation,

$$\left\langle \frac{\partial}{\partial t}(\mathbf{\Omega}^d + \epsilon \mathbf{u}\mathbf{v}^\top) \right\rangle = \eta \mathbf{u}\mathbf{v}^\top \quad \text{s.t.} \quad \eta > 0.$$

The perturbation $\epsilon \mathbf{u}\mathbf{v}^\top$ has very simple consequences on the neural dynamics: it projects neural activity from the direction $\mathbf{v}$ onto a direction $\mathbf{u}$ along which the population was silent. In other words, this specific perturbation acts like an additional input that projects into a yet completely uncovered subspace of the neural dynamics.

How does the network dynamics, as described by $\mathbf{C}_r = \left\langle \mathbf{r}_t \mathbf{r}_t^\top \right\rangle$, $\mathbf{M} \propto \left\langle \mathbf{r}_{t-1} \mathbf{r}_t^\top \right\rangle$ and $\mathbf{D} \propto \left\langle \mathbf{x}_t \mathbf{r}_t^\top \right\rangle$, change? Notice that in the standard autoencoder the fast recurrent connections scale with the covariance of the input and the neural responses. If both are on the order of $\mathcal{O}(\epsilon)$, then the change $\delta\mathbf{\Omega}^f$ in

the fast synaptic weights is on the order of $\mathcal{O}(\epsilon^2)$. From the equilibrium state (S.9) we can compute the change $\delta\mathbf{r}_t$ of the neural responses,

$$
\begin{aligned}
\mathbf{r}_t + \delta\mathbf{r}_t &= (\mathbf{\Omega}^f + \delta\mathbf{\Omega}^f + \mu\mathbf{I})^{-1}\left(\mathbf{Fx}_t + (\mathbf{\Omega}^d + \epsilon\mathbf{uv}^\top)(\mathbf{r}_{t-1} + \delta\mathbf{r}_{t-1})\right), \\
&= \left[\mathbf{Q} + \mathbf{Q}\delta\mathbf{\Omega}^f\mathbf{Q}\right]\left(\mathbf{Fx}_t + (\mathbf{\Omega}^d + \epsilon\mathbf{uv}^\top)(\mathbf{r}_{t-1} + \delta\mathbf{r}_{t-1})\right) + \mathcal{O}(\epsilon^2), \\
&= \mathbf{Q}\left(\mathbf{Fx}_t + \mathbf{\Omega}^d\mathbf{r}_{t-1} + (\mathbf{\Omega}^d + \epsilon\mathbf{uv}^\top)\mathbf{r}_{t-1} + \mathbf{\Omega}^d\delta\mathbf{r}_{t-1})\right) + \mathcal{O}(\epsilon^2), \\
&= \mathbf{r}_t + \epsilon\mathbf{Quv}^\top\mathbf{r}_{t-1} + \mathbf{Q}\mathbf{\Omega}^d\delta\mathbf{r}_{t-1} + \mathcal{O}(\epsilon^2).
\end{aligned}
$$

The dynamics of the rates are thus very simple, as one can see by projecting on $\mathbf{u}$ and a random orthogonal direction $\mathbf{u}_\perp$,

$$
\mathbf{u}^\top\delta\mathbf{r}_t \propto \mathbf{v}^\top\mathbf{r}_{t-1},
$$

and

$$
\mathbf{u}_\perp^\top\delta\mathbf{r}_t \propto \mathbf{u}_\perp^\top\mathbf{Q}\mathbf{\Omega}^d\delta\mathbf{r}_{t-1} = \mu^{-1}\mathbf{u}_\perp^\top\mathbf{\Omega}^d\delta\mathbf{r}_{t-1} \quad\Rightarrow\quad \mathbf{u}_\perp^\top\mathbf{r}_{t-1} = 0.
$$

The last assertion comes from the fact that the network dynamics are stable, and thus any perturbations will decay (as there are no inputs driving the perturbation). In summary, the neural responses are unchanged except along $\mathbf{u}$. Therefore,

$$
\mathbf{C}_r \to \mathbf{C}_r + \epsilon^2\gamma^2\mathbf{uu}^\top,
$$
$$
\mathbf{M} \to \mathbf{M} + \epsilon\gamma\mathbf{vu}^\top,
$$

where $\gamma > 0$ represents the variance (S.12) projected from $\mathbf{v}$. This allows us to solve explicitly for the change in the slow synaptic weights using (32),

$$
\left\langle\frac{\partial}{\partial t}(\mathbf{\Omega}^d + \epsilon\mathbf{uv}^\top)\right\rangle \propto \left\langle(\mathbf{r}_t + \delta\mathbf{r}_t - \alpha(\mathbf{\Omega}^d + \epsilon\mathbf{uv}^\top)(\mathbf{r}_{t-1} + \delta\mathbf{r}_{t-1}))(\mathbf{r}_{t-1} + \delta\mathbf{r}_{t-1})^\top\right\rangle,
$$

$$
\begin{aligned}
&= \alpha\mathbf{M}^\top + \epsilon\gamma\mathbf{uv}^\top - \alpha(\mathbf{\Omega}^d + \epsilon\mathbf{uv}^\top)(\mathbf{C}_r + \epsilon^2\gamma^2\mathbf{uu}^\top), \\
&= \alpha\mathbf{M}^\top + \epsilon\gamma\mathbf{uv}^\top - \alpha(\mathbf{\Omega}^d + \epsilon\mathbf{uv}^\top)\mathbf{C}_r + \mathcal{O}(\epsilon^2).
\end{aligned}
$$

Noticing from (S.8) that $\alpha\mathbf{M}^\top = \alpha\mathbf{\Omega}^d\mathbf{C}_r$ (since $\mathbf{\Omega}^d$ was at a fixed point), this expression simplifies to

$$
\begin{aligned}
&= \epsilon\gamma\mathbf{uv}^\top - \alpha\epsilon\mathbf{uv}^\top\mathbf{C}_r, \\
&= \epsilon\gamma\mathbf{uv}^\top - \alpha\epsilon\gamma\mathbf{uv}^\top, \\
&= (1 - \alpha)\epsilon\gamma\mathbf{uv}^\top.
\end{aligned}
$$

For $\alpha < 1$ the proportionality factor $(1 - \alpha)\epsilon\gamma > 0$ is always positive, and so the weights grow along the perturbation.