[Reviews · NeurIPS 2014]

Submitted by Assigned_Reviewer_17

This submission describes a novel autoencoder method, that uses unsupervised learning to configure a recurrent network to encode both the current and past states of an input. I am not a mathematician nor machine learning expert, and thus am not qualified to review the work for technical merit. However, I have extensive experience in neural network modeling, and thus appreciate both the objective and purported accomplishments: the ability to train a recurrent network to store input sequences in an efficient manner using non-supervised learning. The authors describe a mechanism that addresses the problem by breaking it into two stages — autoencoding, and then optimization — that are carried out over different times scales. The assumptions are reasonable and, of the derivations I could follow, they seemed reasonable. This approach is clever and, to the best of my knowledge, is novel. However, I would like to have seen more extensive evaluation of the network performance, including a benchmark comparison with other standard approaches to the problem using supervised learning (e.g., simple recurrent networks; back propagation through time). While I think such results would strengthen the submission, and its appeal to the audience at NIPS, the results strike me as being of sufficient interest and potential importance to be accepted for presentation.
Summary: Interesting approach to the problem of autoencoding, that uses an unsupervised learning method, and preserves sequence information. Seems novel and of potential interest to NIPS audience, however it could be improved by benchmarking results against standard supervised approaches.

Submitted by Assigned_Reviewer_19

The authors have developed a neural network model that can learn to represent the present and past inputs efficiently. By considering the autoencoder network, they have developed a theory to derive a local learning rule. Subsequently, the theory was extended to the problem of the non-whitened input and that of representing memory.
The topic of the paper is interesting and the theory is clearly explained. Although a simulation example is demonstrated, the validity of the assumptions in their theory is unclear.

Specific comment: Two recurrent connections, ¥Omega^d and ¥Omega^f, are necessary to represent the past inputs in the neuronal network. How to interpret these connections physiologically?
Summary: The authors have developed a neural network model that can learn to represent the present and past inputs efficiently. Although only an example is demonstrated, the topic of the paper is interesting and the theory is clearly explained.

Submitted by Assigned_Reviewer_37

This paper proposes a framework of unsupervised learning for recurrent networks. It proposes different temporal learning scales for simultaneously updating the feedforward, recurrent and delayed recurrent connections. The paper clearly presents its main ideas. To my best knowledge, the paper is new and provides an insight of how to learn the different connections in recurrent networks. I believe this work will reach a wide audience.

Suggestions:
Although I believe this learning paradigm can be generalized into spiking neuron networks, the paper will be more persuasive if the author presented the results of spiking network. And, the authors may consider present more detailed simulation results to strengthen this paper.

Minor:
1. line 98: D^T \hat{x} --> \hat{x}
2. line 181: presynaptic firing rates Fx, and postsynaptic input signal r
3. line 317: a semi-colon is missing: D' = (D; \gamma M)
4. The discussion part seems unfinished
Summary: This work is compelling and technically sound. The manuscript is clearly written.
Author Feedback
Author rebuttal: We appreciate the thoughtful comments and suggestions of the reviewers. It has been mentioned several times that the simulation results should be strengthened and we are planning to update this section in several ways. First, we will simulate larger networks with higher dimensional inputs. Second, we will include more detailed metrics on the firing rate statistics and the reconstruction error over learning, and display the input filter (before/after) as well as a representation of optimal vs learned weights. Third, we will report the performance (and its variations) over several runs from different initial conditions to highlight the stability of learning rules. While we would love to include simulations of the spiking network, we feel that we need the space to clearly describe and lay out the ideas behind the derivation of the learning rules. A spiking network simulation would require a much more in-depth section about the spiking network solution.

# Reply Ref 1

1. [...] including a benchmark comparison with other standard approaches to the problem using supervised learning (e.g., simple recurrent networks; backpropagation through time).

We generally agree that new methods should be evaluated against competing algorithms. We skipped this comparison here because none of the competing algorithms needs to work under the severe constraints of locality and virtually all of them should perform better. Nonetheless, note that, in the trained network, 10 neurons can perfectly represent 8 - 9 inputs. The absolute maximum is 10 (the maximum number of degrees of freedom of the network), so the performance is close to the optimum.

# Reply Ref 2

1. Two recurrent connections, Omega^d and Omega^f, are necessary to represent the past inputs in the neuronal network. How to interpret these connections physiologically?

The time-scale of synaptic transmission is modulated both by the neurotransmitter (e.g. Goaillard, Taylor, Shulz, Marder, Nature Neuroscience 12, 1424 - 1430 (2009)) as well as by the position of the synaptic vesicle in the dendritic tree. As a possible implementation, fast and slow synaptic transmission can be accomplished by proximal and distal dendritic inputs. We are currently working on the implementation of biophysical features like Dale’s law and will investigate different physiological mechanisms.

# Reply Ref 3

We hope we addressed your concerns about the simulation at the beginning of this reply.